# Relationships between the Structural Characteristics of General Medical Practices and the Socioeconomic Status of Patients with Diabetes-Related Performance Indicators in Primary Care

**DOI:** 10.3390/healthcare12070704

**Published:** 2024-03-22

**Authors:** Undraa Jargalsaikhan, Feras Kasabji, Ferenc Vincze, Anita Pálinkás, László Kőrösi, János Sándor

**Affiliations:** 1Department of Public Health and Epidemiology, Faculty of Medicine, University of Debrecen, H-4012 Debrecen, Hungary; jargalsaikhan.undraa@med.unideb.hu (U.J.); kasabji.feras@med.unideb.hu (F.K.); vincze.ferenc@med.unideb.hu (F.V.); palinkas.anita@med.unideb.hu (A.P.); 2Doctoral School of Health Sciences, University of Debrecen, H-4012 Debrecen, Hungary; 3Department of Financing, National Health Insurance Fund, H-1139 Budapest, Hungary; korosi.l@neak.gov.hu; 4HUN-REN-DE Public Health Research Group, Department of Public Health and Epidemiology, Faculty of Medicine, University of Debrecen, H-4012 Debrecen, Hungary

**Keywords:** primary care, diabetes mellitus, performance indicators, patient characteristics, general medical practice characteristics, monitoring

## Abstract

The implementation of monitoring for general medical practice (GMP) can contribute to improving the quality of diabetes mellitus (DM) care. Our study aimed to describe the associations of DM care performance indicators with the structural characteristics of GMPs and the socioeconomic status (SES) of patients. Using data from 2018 covering the whole country, GMP-specific indicators standardized by patient age, sex, and eligibility for exemption certificates were computed for adults. Linear regression models were applied to evaluate the relationships between GMP-specific parameters (list size, residence type, geographical location, general practitioner (GP) vacancy and their age) and patient SES (education, employment, proportion of Roma adults, housing density) and DM care indicators. Patients received 58.64% of the required medical interventions. A lower level of education (hemoglobin A1c test: *β* = −0.108; ophthalmic examination: *β* = −0.100; serum creatinine test: *β =* −0.103; and serum lipid status test: *β* = −0.108) and large GMP size (hemoglobin A1c test: *β =* −0.068; ophthalmological examination *β* = −0.031; serum creatinine measurement *β* = −0.053; influenza immunization *β* = −0.040; and serum lipid status test *β* = −0.068) were associated with poor indicators. A GP age older than 65 years was associated with lower indicators (hemoglobin A1c test: *β* = −0.082; serum creatinine measurement: *β* = −0.086; serum lipid status test: *β* = −0.082; and influenza immunization: *β* = −0.032). Overall, the GMP-level DM care indicators were significantly influenced by GMP characteristics and patient SES. Therefore, proper diabetes care monitoring for the personal achievements of GPs should involve the application of adjusted performance indicators.

## 1. Introduction

According to the International Diabetes Federation, 537 million people were diagnosed with diabetes mellitus (DM) worldwide in 2021, and this figure is expected to increase to 643 million by 2030 [1]. DM affected 9.1% of Hungary’s population in 2021, representing 661,400 people suffering from this disease [1]. DM can effectuate micro- and macrovascular complications, such as late-stage renal failure, blindness, neuropathy, lower-limb amputation, stroke, and myocardial infarction [2], imposing enormous social and economic burdens beyond direct health losses. DM care is mainly provided by primary health care providers [3], and patients achieve better health outcomes with strong primary care [4]. This is mainly due to the efficient delivery of high-quality care with reduced inequalities. One effective management method is glycemic control, as it reduces the risk of retinopathy and nephropathy [5,6]. Similarly, stringent blood pressure control potentially reduces the mortality rate of DM-related complications [7].

In England, the pay-for-performance (P4P) program has been implemented as a strategic instrument for improving health care quality in several countries. The National Health Service’s (NHS) Quality and Outcomes Framework (QOF) is one of the best P4P systems [8,9]. The QOF developed crude clinical indicators, created by the National Institute for Health and Care Excellence (NICE), based on evidence-based indicators for DM care. Additionally, the QOF utilizes financial incentives to encourage adherence to increase the possibility of thriving worldwide [10,11,12]. DM performance indicators are designed to identify areas for advancement, establishing priorities for higher quality, general support, development and benchmark performance, and monitoring [11,12,13]. Moreover, increasing the quality of treatment has improved DM management [14,15]. Additionally, the Carr—Hill allocation formula was introduced to ensure a smooth fund flow for patients’ special needs [16]. The European Best Information through Regional Outcomes in Diabetes (EUBIROD) project included 53 comprehensive indicators of DM care and measured quality of care and outcomes [17,18].

The National Health Insurance Fund (NHIF) in Hungary established a nationally integrated system of GMP-level indicators and a P4P system. This monitoring tool consists of only two DM indicators, hemoglobin A1C testing and ophthalmological investigation frequency, which are assessed at the GMP level [12]. The NHIF is the only entity in Hungary that covers the whole country. The overwhelming majority of Hungarian GPs work in their own solo practice with one nurse. Their service is free of charge for patients. Each GMP is monitored for performance by the NHIF. DM monitoring at the level of primary care is of high importance in Hungary because DM care is mainly managed by the GP from the screening and diagnosis, through the supporting of lifestyle rationalization, to the reference of patients for laboratory and clinical investigations.

According to a recent investigation, DM care in Hungary has improved in respect of glycemic control (11.53%), diastolic blood pressure (6.26%), and total cholesterol (6.06%) therapeutic target achievement. This improvement was more pronounced among patients with a higher level of education [19]. On the other hand, overall primary care was classified as weak in the European setting with respect to governance, coordination, and comprehensiveness in Hungary [20]. However, no studies have been conducted at the national level to investigate the quality of DM care by an extended set of indicators. This area of high concern has been continually ignored, and an extended set of indicators is not computed regularly for routine statistics. It can be inferred that GMP-level monitoring in Hungary is far from successful and cannot contribute properly to improving the quality of DM care.

The aims of this study were (1) to establish an extended GMP-level indicator set for DM care quality in Hungary and (2) to determine associations of indicators with the characteristics of GMPs in order (3) to formulate recommendations for P4P scheme improvement.

## 2. Materials and Methods

### 2.1. Settings

This study was a secondary database analysis. The database built up in the NHIF to evaluate the achievements of a primary care development pilot program implemented from 2012 to 2017 was utilized in our analysis [21,22].

DM patients were defined using the operational definition of the NHIF by at least 4 redemptions of DM medicines in a year. In our cross-sectional study, all GMPs in Hungary (4784 GMPs) providing care for adults exclusively or for adults and children were studied from April 2017 to March 2018. The data required for assessing indicators (patient characteristics and GMP structural parameters) were retrieved from the NHIF’s integrated information system. NHIF is the national institution that contracts with each GMP of Hungary. The Hungarian Central Statistical Office (HCSO) provided the SES data from the 2011 Hungarian Census (the last census before the study period).

### 2.2. Explanatory Variables

GMP structural characteristics were determined by the number of patients receiving GMP care (list size categories: ≤800, 801–1200, 1201–1600, 1601–2000, and ≥2001), status of GP vacancy (distinguishing GMPs managed by temporary GPs with limited time availability and permanent positions with contracted GPs with continuous availability), type of residence (urban, rural), age of GP (<65 or 65 ≥ years), type of GMP by patients provided (making a distinction between GMPs for adults only and GMPs both for adults and for children as mixed), and the county of GMPs.

The GMP-level SES indicators were based on census data as the Carstairs index and the Townsend deprivation scores were computed in the UK [23,24].

The number of years of school attendance among inhabitants aged at least 7 years was determined for each type of residence. The expected number of years of school attendance was calculated using national age- and sex-specific reference values and the demographic data of the residential areas. The indirectly standardized age- and sex-adjusted relative level of education (srEDU) were computed for areas of residence.

The settlement-specific employment ratio was standardized by age and sex to compute the standardized relative employment ratio (srEMP). The expected numbers for each area of residence were calculated using the age- and sex-specific national reference employment ratios (over the age of 15) and settlement data by demographic stratum.

The crowding index for each area of residence was calculated as the number of occupants over the number of rooms in the area. The average Hungarian crowding index was used to determine the residence-specific relative housing density (rHD).

The only significant ethnic minority of Hungary is the Roma. The relative Roma proportion (rRP) for an area of residence was defined as the proportion of self-declared Roma individuals in an area divided by the population share of self-declared Roma individuals in Hungary.

Some GMPs provide care for adults living in different areas. Therefore, the weighted average of the area-specific SES indicators was computed to determine the GMP-specific standardized SES indicators using the area of residence distribution of adults belonging to a GMP. The GMP-specific SES indicators were sorted into tertiles.

### 2.3. Outcome Variables: DM Care Quality Indicators

The EUBIROD process indicators were taken into consideration [17,18]. Because data for microalbuminuria and blood pressure assessment and foot examinations are not registered properly in the NHIF system, a restricted set of indicators could be investigated. The seven (two a part of the routine monitoring system, five newly developed based on the data availability in NHIF) indicators assessed were the proportion of DM patients who underwent hemoglobin A1c testing, ophthalmological examination, serum creatinine measurement, and serum lipid status measurement; the prevalence of DM patients among individuals aged 40–54 years and among individuals aged 55–69 years (the same age groups are used in the NHIF P4P system to monitor hypertension prevalence); and the proportion of diabetes patients under 65 years of age who were vaccinated against influenza (Table 1).

Because the sociodemographic status of patients influences the equality of DM care [12], GMP-specific DM care indicators were indirectly standardized by age group (18–19, 20–24, 25–29, 30–34, 35–39, 40–44, 45–49, 50–54, 55–59, 60–64, 70–74, 75–79, 80–84, 85–89, and 90≤), sex, and eligibility for exemption certificates (issued by the local municipalities for patients with disadvantaged social status and with chronic disease to ensure free-of-charge access to medicines). The national stratum-specific reference values and the demographic composition of the population provided by a GMP resulted in the expected number of GMPs. The indirect standardized ratios (ISRs) were calculated for each indicator and for each GMP.

### 2.4. Statistical Analysis

The ISRs were transformed by an empirical Bayes adjustment to address the problem of the low number of observed cases [25]. Then, the empirical Bayes-adjusted ISRs were normalized (nISRs) using the two-step Box–Cox method [26,27]. Multivariable linear regression models were applied to evaluate the relationship between GMP characteristics and transformed diabetes indicators (nISRs). Standardized linear regression coefficients (β) and their corresponding 95% confidence intervals (95% CI) were calculated. PASW Statistics (version 18.0, SPSS Inc., Chicago, IL, USA) was used for the analysis.

## 3. Results

### 3.1. Descriptive Statistics

In this study, 4784 GMPs and 516,052 DM patients were investigated. The mean (± standard deviation, SD) age of the DM patients was 65.73 (±12.37). The male/female ratio was 1.11. An exemption certificate was issued for 8.34% of the DM patients (n = 43,064). The average length of education among at least 7-years-old persons was 10.7 years. A total of 46.44% of the 15-year-old population was employed. The average number of people occupying one room was 1.08. A total of 3.10% of the population self-reported being Roma. Most of the studied GMPs were rural (66.3%), and the proportion of GMPs with vacant GP positions was 3.70%. The most common category by list size was 1201–1600 patients (32.0%). The majority (77.89%) of the GPs were less than 65 years (Table 2).

The prevalence of DM was 3.33% among adults aged 40–54 years (67,942 DM patients/2,042,573 adults) and 12.65% among those aged 55–69 years (225,816 DM patients/1,785,514 adults). Furthermore, in 2018, our study revealed that 58.64% of the DM patients required medical interventions. A total of 86.18% of patients had serum creatinine testing, while only 12.89% had received an influenza vaccine. Hemoglobin A1c testing, an ophthalmological examination, and blood fat testing were performed on 78.05%, 38.03%, and 78.05% of patients, respectively (Detailed descriptive statistics are summarized in Appendix A Table A1).

The distributions of the nISRs are presented as histograms in Appendix A (Figure A1a–g).

According to the bivariate analyzes, each indicator was reduced in GMPs with a GP aged over 65. The process indicators were in an inverse relationship with the list size. Similar associations were observed for a low level of education and for rurality (apart from the influenza vaccination coverage). The prevalence of DM was lower among patients with an urban residential place, higher level of education, being employed, and living in less crowded households. (Appendix A Table A2).

### 3.2. Linear Regression Modeling

The regression models demonstrated that both patient and GMP characteristics had significant impacts on each indicator (Table 3).

According to the standardized regression coefficients, a low level of education was the strongest determinant for each process indicator (*β_HbA1c test_* = −0.108, 95% CI = −0.153 to −0.063; *β_ophthalmolgic examination_* = −0.100, 95% CI = −0.142 to −0.057; *β_serum creatinine test_* = −0.103, 95% CI = −0.148 to −0.058; *β_serum lipid testing_* = −0.108, 95% CI = −0.153 to −0.063) in addition to influenza vaccination, and a high level was the strongest determinant for both prevalence indicators (*β_40–54 years_* = −0.176, 95% CI = −0.222 to −0.131; *β_55–69 years_* = −0.139, 95% CI = −0.185 to −0.093).

The second most important determinant of the process indicators was a GP age older than 65 years (*β_HbA1c test_* = −0.082, 95% CI = −0.110 to −0.055; *β_serum creatinine test_* = −0.086, 95% CI = −0.113 to −0.058; and *β_serum lipid test_* = −0.082, 95% CI= −0.110 to −0.055).

A higher employment ratio seemed to protect against DM prevalence in both studied age groups (*β_40–54 years_* = −0.067, 95% CI: −0.115 to −0.020; *β_55–69 years_* = −0.111, 95% CI: −0.159 to −0.063). In the case of influenza vaccination, the employment ratio (*β_low/medium_* = 0.056, 95% CI: 0.008 to 0.104) and the GMP type (*β_adult/mixed_* = −0.057, 95% CI: −0.107 to −0.007) were the two strongest risk factors.

A higher prevalence of DM was associated with a higher housing density. Housing density was the second strongest determinant of an ophthalmologic examination (*β_low/medium_* = −0.084, 95% CI: −0.126 to −0.043), but its effect was not proportional (*β_high/medium_* = −0.038, 95% CI: −0.074 to −0.002).

The only observed significant impact of GP vacancy and the proportion of Roma individuals was a decrease in influenza vaccination (*β_GP vacancy_* = −0.053, 95% CI: −0.083 to −0.022; *β_low Roma proportion_* = 0.051, 95% CI: −0.006 to 0.096).

Urbanization had a similar but weaker effect on each studied indicator than education level. Our models demonstrated the worsening impact of a large list size on the process indicators. Among the adult GMPs, more laboratory investigation-based process indicators (HbA1c, serum creatinine, and lipid status) were detected than among the mixed GMPs.

The geographical inequality described by the regression models was large for each DM care indicator. (All the regression models with county-specific regression coefficients are presented in Appendix A Table A3).

## 4. Discussion

### 4.1. Main Findings

Our investigation demonstrated that it is feasible to produce a set of seven indicators for DM care in the Hungarian setting using only the available data provided by the NHIF without targeted primary data collection. Because the templates used to define these indicators were obtained from the NHS QOF and EUBIROD recommendations, the resulting indicators met the general requirements [28].

According to the crude indicators, the performance of Hungarian GMPs was far from the recommendation, and the prevalence of DM was less than the global reference [29]. The GMP-level indicators showed a wider variability for preventive services (the SD of the nISR was 0.428 for influenza vaccination and 0.364 for the prevalence of DM among 40- to 54-year-old adults). The GMP-level nISRs exhibited the narrowest distribution for the glycemic status evaluation indicator (the SD of the nISR was 0.082 for the HbA1c measurement), which is the most important laboratory examination in DM care, followed by the lipid and serum creatinine levels (the SD of the nISR was 0.133 for lipids and 0.133 for serum creatinine).

The regression models revealed that GMP performance was determined by factors unrelated to GMP staff. The proxy measures showed that the SES of patients (mainly education) had a greater impact than GMP structural parameters.

Our findings demonstrated that patients’ education as a proxy measure of SES deprivation and the age of GPs had a significant impact on the outcome indicators similarly seen in other publications [30,31,32,33,34]. The lower likelihood of receiving lipid profiles, eye examinations, microalbumin screening, aspirin therapy, and vaccinations in rural practices is in concordance with other countries’ experiences [35,36]. The wide geographical variability of DM care quality we could observe is also well demonstrated in other countries [37,38]. Furthermore, the presence of a GP vacancy is recognized as a risk factor affecting the quality of DM care [39], as it was demonstrated in our study for influenza vaccination. The higher prevalence of DM among disadvantaged social groups observed in our study is also consistent with publications from various European countries [40,41]. The well-known risk of low-quality DM care among ethnic and racial minorities was particularly evident in our study in the context of the influenza vaccination coverage among Roma [42,43,44].

Internationally, the relationship between list size and process indicators is not well defined [45,46]. On the other hand, our study demonstrated that in Hungary, a bigger list size diminishes GMP performance. The explanation for the published variability and the background of the Hungarian findings needs further investigations. Similarly, the statistically significant improvement in the performance of GMPs providing care to adults only, in terms of hemoglobinA1c testing, serum creatinine determination, and serum lipid status testing, along with the significant higher influenza vaccination coverage in GMPs providing care for both adults and children, requires further explorative investigation in the future.

The research suggests that a higher employment among patients in a community can contribute to a reduced focus on delivering optimal DM care to them [39,47], which is in accordance with our study results, which demonstrated an inverse association between influenza vaccination coverage and unemployment. Furthermore, living conditions, exemplified by overcrowding in our study, have an impact on the quality of DM care [42,48]. The Hungarian observation was a significant but not monotonic relationship between overcrowding and an ophthalmological examination.

Considering the strong negative impact of a low level of education, large list size, the higher age of GPs, the rural location (and of certain counties), if the crude GMP indicators are used to measure the GMP-staffs’ personal contribution, then the GMP-staff performances are underestimated in GMPs with disadvantaged conditions. Because the P4P system is, primarily, to facilitate the better performance of GMP staffs, P4P is obviously counterproductive without an adjustment for GMP-staff-independent factors.

### 4.2. Strengths and Limitations

The main strength of this research was the inclusion of all Hungarian GMPs and the whole adult population of Hungary, which prevented selection bias and ensured proper statistical power for analysis. Furthermore, misclassification of the studied parameters was avoided due to the standardized data collection of the NHIF and the HCSO.

However, we could not account for the additional characteristics of patients that may have influenced the quality of DM care, such as other social determinants [49], health literacy level [50], and health behaviors [49,51]. Further studies are needed to evaluate the inequalities in other social dimensions. The SES indicators were computed from data from 2011, while the GMP structural parameters and the DM indicators were computed from data from 2018. Because change in SES is a slow process, the 7-year gap between these two datasets may have influenced the observed results only to a small degree. However, this bias may have resulted in weakened observed associations.

Because the NHIF has no data on the results of the DM-care-related investigations, the short-term outcome indicators could not be analyzed in our investigation.

Finally, the mechanisms behind the observed associations between the GMP structural characteristics and DM care quality could not be investigated in our cross-sectional investigation. More studies are needed to utilize the opportunities to improve DM care by improving geographical availability in rural areas, by providing care more adapted to less educated patients, by supporting the GMPs with a bigger list size, and by understanding the negative impact of a GP age over 65 on process indicators.

### 4.3. Implications

The quality of DM care is dependent both on patients’ SES and on the structure of GMPs. Because the indicators in the P4P system are used to improve GMP-staffs’ personal performance, they must be adjusted to remove the impact of factors that are independent and not influenceable by GPs and nurses. Obviously, the more comprehensive the adjustment is, the more sensitive the indicator is to the personal performance of the GMP staff, and consequently, the more effective the monitoring of malpractice identification and benchmarking.

On the other hand, problems in DM care identified by indicators adjusted for certain factors can no longer be identified by these adjusted indicators. Consequently, performance monitoring must use both the crude and the adjusted indicators in parallel.

Considering that 58.64% of required medical interventions are provided according to the summarized process indicator and that there is substantial variability in each indicator observed in Hungary, there is an explicit need for better monitoring, which could motivate GPs and nurses to improve their personal performance and establish interventions targeting GMP-staff-unrelated quality determinants.

## 5. Conclusions

Altogether, (1) seven standardized indicators for DM care were successfully elaborated for Hungarian GMP monitoring. (2) These indicators showed a significant deviation in the average performance of the relevant recommendations and a large variability in GMP performance. SES and structural factors explain GMP-level variability and determine the quality of DM care. (3) To enhance the effectiveness of the P4P scheme, a broader range of indicators, specifically addressing the personal contributions of GMP-staff members to the quality of care, is necessary. The nonadjusted indicators must also be applied to monitor DM care quality from the patient’s perspective.

## Figures and Tables

**Table 1 healthcare-12-00704-t001:** Definitions of the quality of DM care indicators.

	Indicator Name	Target Group	Indicator Definition
Process indicators	HemoglobinA1c testing	Primary health care patients with DM	Proportion of diabetics who were involved in hemoglobin A1c testing (at least once in previous 12 months)
	Ophthalmological examination	Primary health care patients with DM	Proportion of diabetics who participated in an ophthalmological examination (at least once in previous 12 months)
	Serum creatinine check	Primary health care patients with DM	Proportion of diabetic patients who participated in a serum creatinine determination (at least once in previous 12 months)
	Influenza vaccination	Primary health care patients with DM under 65 years old	Proportion of diabetic patients under 65 years of age who were vaccinated against influenza
	Lipid status checking	Primary health care patients with DM	Proportion of diabetics who participated in a serum lipid status test (at least once in previous 12 months)
Prevalence indicators	DM patients aged 40–54 years	40–54-years-old primary health care patients with DM	Proportion of diabetic patients, aged 40–54 years, who redeemed a diabetic medicine at least 4 times in the previous 12 months
	DM patients aged 55–69 years	55–69-years-old primary health care patient with DM	Proportion of diabetic patients, aged 55–69 years, who redeemed a diabetic medicine at least 4 times in the previous 12 months

**Table 2 healthcare-12-00704-t002:** Characteristics of the investigated general medical practices (GMPs) and patients with diabetes mellitus (DM).

Structural Characteristics of the GMP	Categories	GMP		DM Patients	
		N	Percentage	N	Percentage
Settlement type	Rural	3172	66.30%	365,381	70.80%
	Urban	1612	33.70%	150,671	29.20%
GMP list size	<800	153	3.00%	7207	1.40%
	801–1200	655	14.00%	45,335	8.78%
	1201–1600	1522	32.00%	142,544	27.62%
	1601–2000	1504	31.00%	178,537	34.60%
	>2000	950	20.00%	142,429	27.60%
GP vacancy	Filled	4608	96.30%	503,713	97.60%
	Vacant	176	3.70%	12,339	2.40%
GMP type	Adult	3317	69.00%	385,372	74.68%
	Mixed	1467	31.00%	130,680	25.32%
Age of GP (years)	<65	1019	22.11%	103,895	20.12%
	≥65	3589	77.89%	399,818	77.48%
County	Baranya	207	4.33%	21,458	4.15%
	Bács-Kiskun	256	5.35%	27,720	5.37%
	Békés	187	3.91%	19,939	3.86%
	Borsod-Abaúj-Zemplén	370	7.73%	35,345	6.85%
	Csongrád	204	4.26%	18,907	3.66%
	Fejér	194	4.06%	22,576	4.37%
	Győr-Moson-Sopron	202	4.22%	23,251	4.51%
	Hajdú-Bihar	242	5.06%	25,993	5.04%
	Heves	160	3.34%	16,616	3.22%
	Komárom-Esztergom	141	2.95%	16,105	3.12%
	Nógrád	109	2.28%	10,537	2.04%
	Pest	466	9.74%	61,273	11.87%
	Somogy	172	3.60%	18,934	3.67%
	Szabolcs-Szatmár-Bereg	265	5.54%	28,431	5.50%
	Jász-Nagykun-Szolnok	192	4.01%	21,118	4.10%
	Tolna	119	2.49%	14,144	2.74%
	Vas	133	2.78%	14,551	2.82%
	Veszprém	164	3.43%	18,889	3.66%
	Zala	141	2.95%	14,868	2.90%
	Budapest	860	17.98%	85,397	16.55%
Total		4784	100%	516,052	100%

**Table 3 healthcare-12-00704-t003:** Associations between the structural characteristics and socioeconomic status indicators of GMPs and standardized DM care indicators according to standardized regression coefficients from multivariable linear regression analyses and their corresponding 95% confidence intervals.

	Process Indicators					DM Prevalence	
	HemoglobinA1c Testing	Ophthalmological Examination	Serum Creatinine Testing	Serum Lipid Status Testing	Influenza Vaccination	Among 40–54-Years-Old	Among 55–69-Years-Old
**GMP characteristics**							
GMP type (adult/mixed)	**0.069 [0.021**; **0.117]**	0.017 [−0.029; 0.062]	**0.084 [0.035**; **0.132]**	**0.069 [0.021**; **0.117]**	**−0.057 [−0.107**; **−0.007]**	−0.025 [−0.072; 0.023]	0.022 [−0.026; 0.070]
Settlement type (urban/rural)	**0.059 [0.010**; **0.108]**	**0.048 [0.001**; **0.094]**	**0.075 [0.026**; **0.125]**	**0.059 [0.010**; **0.108]**	−0.007 [−0.058; 0.045]	**−0.058 [−0.107**; **−0.009]**	**−0.072 [−0.121**; **−0.023]**
GP (vacancy/ age < 65)	0.000 [−0.029; 0.029]	0.013 [−0.014; 0.041]	0.014 [−0.015; 0.044]	0.000 [−0.029, 0.029]	**−0.053 [−0.083**; **−0.022]**	0.012 [−0.017; 0.041]	0.000 [−0.030; 0.029]
GP (age ≥ 65X/age < 65)	**−0.082 [−0.110**; **−0.055]**	−0.025 [−0.051; 0.001]	**−0.086 [−0.113**; **−0.058]**	**−0.082 [−0.110**; **−0.055]**	**−0.032 [−0.060**; **−0.003]**	**−0.036 [−0.063**; **−0.009]**	**−0.050 [−0.078**; **−0.023]**
List size (≤800/1201–1600)	−0.009 [−0.038; 0.021]	**−0.037 [−0.064**; **−0.009]**	−0.007 [−0.037; 0.022]	−0.008 [−0.038; 0.021]	0.009 [−0.021; 0.040]	−0.017 [−0.046; 0.012]	−0.017 [−0.047; 0.012]
List size (801–1200/1201–1600)	−0.019 [−0.049; 0.011]	−0.001 [−0.029; 0.028]	−0.017 [−0.047; 0.014]	−0.019 [−0.049; 0.011]	0.003 [−0.028; 0.035]	−0.021 [−0.051; 0.009]	−0.015 [−0.045; 0.015]
List size (1601–2000/1201–1600)	−0.017 [−0.048; 0.015]	**−0.031 [−0.061**; **−0.001]**	−0.001 [−0.033; 0.030]	−0.017 [−0.048; 0.015]	−0.016 [−0.049; 0.017]	0.014 [−0.017; 0.046]	0.016 [−0.016; 0.047]
List size (>2000/1201–1600)	**−0.068 [−0.100**; **−0.036]**	**−0.031 [−0.061**; **−0.001]**	**−0.053 [−0.085**; **−0.021]**	**−0.068 [−0.100**; **−0.036]**	**−0.040 [−0.074**; **−0.007]**	0.003 [−0.029; 0.035]	0.006 [−0.026; 0.038]
**Patient characteristics**							
Level of education (low/medium)	**−0.108 [−0.153**; **−0.063]**	**−0.100 [−0.142**; **−0.057]**	**−0.103 [−0.148**; **−0.058]**	**−0.108 [−0.153**; **−0.063]**	−0.005 [−0.051; 0.042]	**0.048 [0.004**; **0.093]**	0.019 [−0.026; 0.063]
Level of education (high/medium)	**0.048 [0.002**; **0.094]**	0.036 [−0.007; 0.079]	0.036 [−0.010; 0.082]	**0.048 [0.002**; **0.094]**	0.039 [−0.009; 0.087]	**−0.176 [−0.222**; **−0.131]**	**−0.139 [−0.185**; **−0.093]**
Employment ratio (low/medium)	−0.018 [−0.064; 0.028]	−0.018 [−0.061; 0.026]	−0.014 [−0.061; 0.032]	−0.018 [−0.064; 0.028]	**0.056 [0.008**; **0.104]**	**0.051 [0.005**; **0.097]**	**0.061 [0.015**; **0.107]**
Employment ratio (high/medium)	0.002 [−0.046; 0.050]	0.030 [−0.016; 0.075]	−0.024 [−0.072; 0.025]	0.002 [−0.046; 0.050]	0.009 [−0.041; 0.059]	**−0.067 [−0.115**; **−0.020]**	**−0.111 [−0.159**; **−0.063]**
Housing density (low/medium)	0.041 [−0.003; 0.084]	**−0.084 [−0.126**; **−0.043]**	0.018 [−0.025; 0.062]	0.040 [−0.003; 0.084]	−0.001 [−0.046; 0.045]	**−0.056 [−0.099**; **−0.013]**	**−0.049 [−0.092**; **−0.005]**
Housing density (high/medium)	−0.008 [−0.046; 0.031]	**−0.038 [−0.074**; **−0.002]**	−0.013 [−0.051; 0.026]	−0.007 [−0.046; 0.031]	0.000 [−0.040; 0.040]	**0.049 [0.011**; **0.086]**	**0.047 [0.009**; **0.085]**
Proportion Roma (low/medium)	−0.012 [−0.055; 0.031]	0.015 [−0.026; 0.055]	0.016 [−0.027; 0.059]	−0.012 [−0.055; 0.031]	**0.051 [0.006**; **0.096]**	−0.040 [−0.083; 0.002]	0.014 [−0.029; 0.057]
Proportion Roma (high/medium)	−0.014 [−0.060; 0.033]	−0.022 [−0.066; 0.022]	0.014 [−0.033; 0.061]	−0.014 [−0.060; 0.033]	−0.031 [−0.079; 0.018]	0.016 [−0.030; 0.063]	0.039 [−0.007; 0.086]

Adjusted standardized linear regression coefficients with 95% confidence intervals. The values are adjusted for counties. All the models are presented in Appendix A Table A3. Significant results are in bold.

## Data Availability

The datasets used and/or analyzed during the current study are available from the corresponding author on reasonable request.

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
