# Peer review of "Relationships between the Structural Characteristics of General Medical Practices and the Socioeconomic Status of Patients with Diabetes-Related Performance Indicators in Primary Care"

_healthcare, 2024, doi:10.3390/healthcare12070704_

Round 1
Reviewer 1 Report
Comments and Suggestions for Authors
The quality of health care and monitoring of people with diabetes is of great interest. However, a retrospective study design is presented. The relationship between the indicators and follow-up results of people with diabetes with the characteristics of the medical teams that care for them is shown, performing statistical tests that would indicate that the age of the medical team influences patient outcomes.
As the authors themselves recognize, there are many other factors that are related to the care of people with chronic diseases. Furthermore, the primary care team (GMP) do not mention the presence of nurses, who in most countries are the key professionals who carry out the monitoring and health education of these people.
In addition, the age of medical professionals analyzed with indicators of quality of care and outcomes, in my opinion, should not be done. Different investigations suggests that age discrimination can have serious consequences. For the same reason, age could be analyzed in groups, also analyzing recently graduated professionals separately.
Other comments that should be considered:
Section 2. Materials and Methods
· age of GP (<65 or 65≤ years): modify
· geographical location of GMPs by county: not well understood
· GMP-specific DM care indicators were indirectly standardized by age group (18–19, 20–24, 25–29, 30–34, 35–39, 40–44, 45–49, 50–54, 55–59, 60–64, 70–74, 75–79, 80–84, 85– 90, and 90<): The reason is not understood, the authors should explain and justify it.
Limitations
They are discussed briefly, they should be expanded considering the design of the study without causality in the findings, only associations should be mentioned.
Author Response
Dear Reviewer,
Thank you very much for the careful review of our manuscript. Please find enclosed the revised version of the manuscript “Relationships between the structural characteristics of general medical practices and the socioeconomic status of patients with diabetes-related performance indicators in primary care” by Undraa Jargalsaikhan, et al.
Each comment and suggestion has been considered. The corresponding changes and refinements made in the revised paper are summarized in our response after considering each of your suggestion. Answers along with the modifications we made are summarized below (comments/questions of Yours are in capitals).
Sincerely yours, Janos Sandor (on behalf of the authors)
Answers/reflections to the comments of Reviewer-1:
1.
THE QUALITY OF HEALTH CARE AND MONITORING OF PEOPLE WITH DIABETES IS OF GREAT INTEREST. HOWEVER, A RETROSPECTIVE STUDY DESIGN IS PRESENTED. THE RELATIONSHIP BETWEEN THE INDICATORS AND FOLLOW-UP RESULTS OF PEOPLE WITH DIABETES WITH THE CHARACTERISTICS OF THE MEDICAL TEAMS THAT CARE FOR THEM IS SHOWN, PERFORMING STATISTICAL TESTS THAT WOULD INDICATE THAT THE AGE OF THE MEDICAL TEAM INFLUENCES PATIENT OUTCOMES.
Thanks for this general comment!
2.
AS THE AUTHORS THEMSELVES RECOGNIZE, THERE ARE MANY OTHER FACTORS THAT ARE RELATED TO THE CARE OF PEOPLE WITH CHRONIC DISEASES.
Thanks for this comment! To emphasize that our study was to demonstrate the existence of social inequalities and not to describe it in a comprehensive manner, the 4.2. Strengths and limitations section was completed:
Original sentence:
However, we could not account for additional characteristics of patients that may have influenced the quality of DM care, such as other social determinants [47], health literacy level [48], and health behaviors [47,49 ].
Modified text:
However, we could not account for additional characteristics of patients that may have influenced the quality of DM care, such as other social determinants [47], health literacy level [48], and health behaviors [47,49 ]. Further studies are needed to evaluate the inequalities in other social dimensions.
3.
FURTHERMORE, THE PRIMARY CARE TEAM (GMP) DO NOT MENTION THE PRESENCE OF NURSES, WHO IN MOST COUNTRIES ARE THE KEY PROFESSIONALS WHO CARRY OUT THE MONITORING AND HEALTH EDUCATION OF THESE PEOPLE.
We agree with you. Nurses play important role in the management of DM patients. Although, nurses are subordinated to the GP, nurses’ competencies are getting wider and wider. SO, GP specific conclusions were replaced with GMP-staff related conclusions throughout the text.
Original sentence:
The regression models revealed that GMP performance was determined by unrelated GP factors.
Modified text:
The regression models revealed that GMP performance was determined by factors unrelated to GMP staff.
Original sentence:
On the other hand, our study demonstrated that in Hungary, bigger list size diminishes GP performance.
Modified text:
On the other hand, our study demonstrated that in Hungary, bigger list size diminishes GMP performance.
Original sentences:
Considering the strong negative impact of low level of education, large list size, the higher age of GPs, the rural location (and of certain counties), if the crude GMP indicators are used to measure the GPs personal contribution then the GP performances are underestimated in GMPs with disadvantaged conditions. Because the P4P system is, primarily, to facilitate the better performance of GPs, the P4P is obviously counterproductive without adjustment for GP-independent factors.
Modified text:
Considering the strong negative impact of low level of education, large list size, the higher age of GPs, the rural location (and of certain counties), if the crude GMP indicators are used to measure the GMP-staffs’ personal contribution then the GMP-staff performances are underestimated in GMPs with disadvantaged conditions. Because the P4P system is, primarily, to facilitate the better performance of GMP-staffs, the P4P is obviously counterproductive without adjustment for GMP-staff-independent factors.
Original sentence:
Because the indicators in the P4P system are used to improve GPs’ personal performance, they must be adjusted to remove the impact of factors that are independent and not influenceable by GPs. Obviously, the more comprehensive the adjustment is, the more sensitive the indicator is to the personal performance of the GP, and consequently, the more effective the monitoring of malpractice identification and benchmarking.
Modified text:
Because the indicators in the P4P system are used to improve GMP-staffs’ personal performance, they must be adjusted to remove the impact of factors that are independent and not influenceable by GPs and nurses. Obviously, the more comprehensive the adjustment is, the more sensitive the indicator is to the personal performance of the GMP-staff, and consequently, the more effective the monitoring of malpractice identification and benchmarking.
Original sentence:
… there is an explicit need for better monitoring, which could motivate GPs to improve their personal performance and establish interventions targeting GP-unrelated quality determinants.
Modified text:
… there is an explicit need for better monitoring, which could motivate GPs and nurses to improve their personal performance and establish interventions targeting GMP-staff-unrelated quality determinants.
4.
IN ADDITION, THE AGE OF MEDICAL PROFESSIONALS ANALYZED WITH INDICATORS OF QUALITY OF CARE AND OUTCOMES, IN MY OPINION, SHOULD NOT BE DONE. DIFFERENT INVESTIGATIONS SUGGEST THAT AGE DISCRIMINATION CAN HAVE SERIOUS CONSEQUENCES. FOR THE SAME REASON, AGE COULD BE ANALYZED IN GROUPS, ALSO ANALYZING RECENTLY GRADUATED PROFESSIONALS SEPARATELY.
The GPs-aging is a well-known phenomenon in the European countries (European Commission. Towards a Job-Rich Recovery; Commission Staff Working Document on an Action Plan for the EU Health Workforce; European Commission: Brussels, Belgium, 2012.; Barriball L., Bremner J., Buchan J., Craveiro I., Dieleman M., Dix O., Dussault G., Jansen C., Kroezen M., Rafferty A.M., et al. Recruitment and Retention of the Health Workforce in Europe. Consumers, Health, Agriculture and Food Executive Agency of the European Commission; Brussels, Belgium: 2015.), and in Hungary (Papp M, KÅ‘rösi L, Sándor J, Nagy C, Juhász A, Ádány R. Workforce crisis in primary healthcare worldwide: Hungarian example in a longitudinal follow-up study. BMJ Open. 2019 Jul 23;9(7):e024957. doi: 10.1136/bmjopen-2018-024957. PMID: 31340955; PMCID: PMC6661691.). The work of older GPs is required nowadays to ensure the proper primary care. The higher the age of GPs, the more important to know whether they are able to maintain the proper quality in their practices? Our former investigations shown that the quality of care was influenced by the GP age(Pálinkás A, Sándor J. Effectiveness of COVID-19 Vaccination in Preventing All-Cause Mortality among Adults during the Third Wave of the Epidemic in Hungary: Nationwide Retrospective Cohort Study. Vaccines (Basel). 2022 Jun 24;10(7):1009. doi: 10.3390/vaccines10071009. PMID: 35891173; PMCID: PMC9319484.; Kovács N, Varga O, Nagy A, Pálinkás A, Sipos V, KÅ‘rösi L, Ádány R, Sándor J. The impact of general practitioners' gender on process indicators in Hungarian primary healthcare: a nation-wide cross-sectional study. BMJ Open. 2019 Sep 6;9(9):e027296. doi: 10.1136/bmjopen-2018-027296. PMID: 31494598; PMCID: PMC6731795.), even this impact was not manifested at the level of mortality (Sándor J, Pálinkás A, Vincze F, Sipos V, Kovács N, Jenei T, Falusi Z, Pál L, KÅ‘rösi L, Papp M, Ádány R. Association between the General Practitioner Workforce Crisis and Premature Mortality in Hungary: Cross-Sectional Evaluation of Health Insurance Data from 2006 to 2014. Int J Environ Res Public Health. 2018 Jul 2;15(7):1388. doi: 10.3390/ijerph15071388. PMID: 30004449; PMCID: PMC6068803.).
The present paper could also confirm that the age of GP is a quality influencing factor.
Because this paper was not focused to the aging GP population, this argumentation was not inserted into the text.
5.
OTHER COMMENTS THAT SHOULD BE CONSIDERED:
MATERIALS AND METHODS
AGE OF GP (<65 OR 65≤ YEARS): MODIFY
Thanks for this comment! Corrected accordingly.
Original text:
age of GP (<65 or 65≥ years)
Modified text:
age of GP (<65 or 65≥ years)
6.
GEOGRAPHICAL LOCATION OF GMPS BY COUNTY: NOT WELL UNDERSTOOD
Thanks for this comment! Corrected accordingly.
Original text:
… geographical location of GMPs by county.
Modified text:
… county of GMPs.
7.
GMP-SPECIFIC DM CARE INDICATORS WERE INDIRECTLY STANDARDIZED BY AGE GROUP (18–19, 20–24, 25–29, 30–34, 35–39, 40–44, 45–49, 50–54, 55–59, 60–64, 70–74, 75–79, 80–84, 85– 90, AND 90<): THE REASON IS NOT UNDERSTOOD, THE AUTHORS SHOULD EXPLAIN AND JUSTIFY IT.
The indicators’ adjustment forage was justified by the age-dependency of the DM care quality (HbA1c checking, ophthalmologic examination) described in our former investigation (Kovács N, Pálinkás A, Sipos V, Nagy A, Harsha N, KÅ‘rösi L, Papp M, Ádány R, Varga O, Sándor J. Factors Associated with Practice-Level Performance Indicators in Primary Health Care in Hungary: A Nationwide Cross-Sectional Study. Int J Environ Res Public Health. 2019 Aug 29;16(17):3153. doi: 10.3390/ijerph16173153. PMID: 31470573; PMCID: PMC6747708.) This paper is cited in the manuscript (reference 12).
To make explicit this reason, the text was modified.
Original text:
GMP-specific DM care indicators were indirectly standardized by age group (18–19, 20–24, 25–29, 30–34, 35–39, 40–44, 45–49, 50–54, 55–59, 60–64, 70–74, 75–79, 80–84, 85–90, and 90<),…
Modified text:
Because the socio-demographic status of patients influences the quality of DM care [12], GMP-specific DM care indicators were indirectly standardized by age group (18–19, 20–24, 25–29, 30–34, 35–39, 40–44, 45–49, 50–54, 55–59, 60–64, 70–74, 75–79, 80–84, 85–90, and 90<),…
8.
LIMITATIONS
THEY ARE DISCUSSED BRIEFLY. THEY SHOULD BE EXPANDED CONSIDERING THE DESIGN OF THE STUDY WITHOUT CAUSALITY IN THE FINDINGS, ONLY ASSOCIATIONS SHOULD BE MENTIONED.
The 4.2. Strengths and limitations sections was completed with a sentence to acknowledge this limitation also:
Finally, the mechanisms behind the observed associations between GMP structural characteristics and DM care quality could not be investigated in our cross-sectional investigation.

Reviewer 2 Report
Comments and Suggestions for Authors
Although the objective of the study, to describe the associations of DM care performance indicators with the structural characteristics of GMPs and the socioeconomic status of patients, may be of interest to the scientific community, the article presents important limitations that condition the publication of the article:
- INTRODUCTION:
o To contextualize the paper well it would be necessary to include information on the impact of socioeconomic status on DM and how are the characteristics of GMPs in Hungary in relation to diabetic patient care.
o Some inaccurate epidemiological data
o Some incomplete objective.
- METHODOLOGY:
o Not well detailed the information of the database from where they obtained the information.
o It is not explained why the study period is 7 years ago.
o The description of the independent variables is not clear.
o The selection of age brackets to consider the prevalence of diabetes is not well described.
o Some errors in the description of the categories of the different variables.
o In the statistical analysis, doubts about the lack of a bivariate analysis and about the regression model used in the multivariate analysis.
- RESULTS
o Inaccuracies in some results.
o It is not easy to interpret the results.
o The results show categories that have not been described in the Methodology section.
- DISCUSSION
o It should be noted as a limitation that only 5 process indicators have been considered to measure the quality of care for diabetic patients, most of them being related to the performance of analytical tests. They should explain why they have not been able to select outcome indicators (% of patients with HBA1C, Blood Pressure or Cholesterol controlled, % of patients with chronic complications...). The selected indicators measure very poorly the quality of care for diabetic patients.
Best regards

Author Response
Dear Reviewer,
Thank you very much for the careful review of our manuscript. Please find enclosed the revised version of the manuscript “Relationships between the structural characteristics of general medical practices and the socioeconomic status of patients with diabetes-related performance indicators in primary care” by Undraa Jargalsaikhan, et al.
Each comment and suggestion has been considered. The corresponding changes and refinements made in the revised paper are summarized in our response after considering each of your suggestion. Answers along with the modifications we made are summarized below (comments/questions of Yours are in capitals).
Sincerely yours, Janos Sandor (on behalf of the authors)
Answers/reflections to the comments of Reviewer-2:
1.
ALTHOUGH THE OBJECTIVE OF THE STUDY, TO DESCRIBE THE ASSOCIATIONS OF DM CARE PERFORMANCE INDICATORS WITH THE STRUCTURAL CHARACTERISTICS OF GMPS AND THE SOCIOECONOMIC STATUS OF PATIENTS, MAY BE OF INTEREST TO THE SCIENTIFIC COMMUNITY, THE ARTICLE PRESENTS IMPORTANT LIMITATIONS THAT CONDITION THE PUBLICATION OF THE ARTICLE:
Thanks for this general comment!
2.
INTRODUCTION:
TO CONTEXTUALIZE THE PAPER WELL IT WOULD BE NECESSARY TO INCLUDE INFORMATION ON THE IMPACT OF SOCIOECONOMIC STATUS ON DM AND HOW ARE THE CHARACTERISTICS OF GMPS IN HUNGARY IN RELATION TO DIABETIC PATIENT CARE.
The introduction has been completed as requested to mention that there is an association between the socioeconomic position of patients and the epidemiological characteristics of DM.
Original text:
According to a recent Hungarian investigation, the quality of DM care improved glycemic control by 11.53%, and there was moderate improvement in diastolic blood pressure (6.26%), and total cholesterol (6.06%) therapeutic target achievement [19].
Modified text:
According to a recent Hungarian investigation, the quality of DM care improved in the respect of glycemic control (11.53%), diastolic blood pressure (6.26%), and total cholesterol (6.06%) therapeutic target achievement. This improvement was affected by the level of patients’ education. [19].
3.
SOME INACCURATE EPIDEMIOLOGICAL DATA
Unfortunately, we could not identify the cause of this criticism.
4.
SOME INCOMPLETE OBJECTIVE.
Unfortunately, we could not identify the cause of this criticism.
5.
METHODOLOGY:
NOT WELL DETAILED THE INFORMATION OF THE DATABASE FROM WHERE THEY OBTAINED THE INFORMATION.
Thanks for this comment! The National Health Insurance Fund is contracted with each GMPs in Hungary, and has data for the operation of each GMPs. To emphasize it the text was corrected.
Original text:
The data required for assessing indicators (patient characteristics and GMP structural parameters) were retrieved from the NHIF's integrated information system.
Modified text:
The data required for assessing indicators (patient characteristics and GMP structural parameters) were retrieved from the NHIF's integrated information system. NHIF is the national institution that contracts with each GMP of Hungary.
6.
IT IS NOT EXPLAINED WHY THE STUDY PERIOD IS 7 YEARS AGO.
The Swiss-Hungarian Cooperation Programme entitled ‘Public Health Focused Model Programme for Organising Primary Care Services Backed by a Virtual Care Service Centre’ was a pilot programme targeting the foundation of the Hungarian PHC reform that encourages an improvement in the general health status of the population, thereby substantially reducing social inequalities in health. This programme was implemented from 2012 to 2017. (Ádány R, Kósa K, Sándor J, Papp M, Fürjes G. General practitioners' cluster: a model to reorient primary health care to public health services. Eur J Public Health. 2013 Aug;23(4):529-30. doi: 10.1093/eurpub/ckt095. PMID: 23882116; Jakab Z. Public health, primary care and the 'cluster' model. Eur J Public Health. 2013 Aug;23(4):528. doi: 10.1093/eurpub/ckt091. Epub 2013 Jun 27. PMID: 23813715.) The achievement of the programme was assessed by a before-after analysis. The database built up in the NHIF to describe the after programme status was utilized in our analysis. Similar database for later period was not available for analysis, unfortunately.
The Setting subsection was completed accordingly.
Original text:
This study was a secondary database analysis.
Modified text:
This study was a secondary database analysis. The database built up in the NHIF to evaluate the achievements of a primary care development pilot program implemented from 2012 to 2017 was utilized in our analysis. [19,20]
The two citations were inserted into the references, and the references were renumbered as necessary.
7.
THE DESCRIPTION OF THE INDEPENDENT VARIABLES IS NOT CLEAR.
Unfortunately, we could not identify the cause of this criticism.
8.
THE SELECTION OF AGE BRACKETS TO CONSIDER THE PREVALENCE OF DIABETES IS NOT WELL DESCRIBED.
If I understand the comment properly, then the definition of age groups applied in DM prevalence calculation needs explanation.
The P4P system operated by the National Health Insurance Fund (mentioned in the Introduction, 3rd paragraph) uses prevalence indicators for hypertension. In this system, the prevalence of hypertension is calculated for two age groups: for younger adults with age of 40-54, and for older adults with age of 55-69. The same age groups were applied in our investigation.
9.
SOME ERRORS IN THE DESCRIPTION OF THE CATEGORIES OF THE DIFFERENT VARIABLES.
Thanks for the careful checking! Corrected accordingly.
Original text in Table 2:
|
≥65 |
|
<65 |
Modified text in Table 2:
|
<65 |
|
≥65 |
10.
IN THE STATISTICAL ANALYSIS, DOUBTS ABOUT THE LACK OF A BIVARIATE ANALYSIS AND ABOUT THE REGRESSION MODEL USED IN THE MULTIVARIATE ANALYSIS.
Thanks for this suggestion!
The bivariate linear regression analyses are summarized in Appendix Table A2, and a sentence was inserted into the end of 3.1. Descriptive Statistics section:
The associations between GMP characteristics and nISRs by bivariate linear regression analyses are shown Appendix Table A2.
Consequently, the numbering of the former Appendix Table A2 had to be changed to Appendix Table A3.
The footnote for the former Appendix TableA2 was corrected.
Original footnote:
*Adjusted standardized linear regression coefficients with their 95% confidence intervals. Values are adjusted for counties. Significant results bolded.
Corrected footnote:
*Adjusted standardized linear regression coefficients with their 95% confidence intervals. Significant results bolded.
11.
RESULTS
INACCURACIES IN SOME RESULTS.
Thanks for the careful checking! Corrected accordingly.
Original text:
The majority (77.89%) of the GPs were older than 65 years.
Modified text:
The majority (77.89%) of the GPs were less than 65 years.
Original text:
… while only 12.86% had received an influenza vaccine.
Modified text:
… while only 12.89% had received an influenza vaccine.
12.
IT IS NOT EASY TO INTERPRET THE RESULTS.
Unfortunately, we could not identify the cause of this criticism.
13.
THE RESULTS SHOW CATEGORIES THAT HAVE NOT BEEN DESCRIBED IN THE METHODOLOGY SECTION.
Unfortunately, we could not identify the cause of this criticism.
14.
IT SHOULD BE NOTED AS A LIMITATION THAT ONLY 5 PROCESS INDICATORS HAVE BEEN CONSIDERED TO MEASURE THE QUALITY OF CARE FOR DIABETIC PATIENTS, MOST OF THEM BEING RELATED TO THE PERFORMANCE OF ANALYTICAL TESTS. THEY SHOULD EXPLAIN WHY THEY HAVE NOT BEEN ABLE TO SELECT OUTCOME INDICATORS (% OF PATIENTS WITH HBA1C, BLOOD PRESSURE OR CHOLESTEROL CONTROLLED, % OF PATIENTS WITH CHRONIC COMPLICATIONS...). THE SELECTED INDICATORS MEASURE VERY POORLY THE QUALITY OF CARE FOR DIABETIC PATIENTS.
Thanks for this comment! It is an obvious limitation of our investigation. To acknowledge it, this sentence was inserted into the 4.2. Strengths and limitations section:
Because the NHIF has no data on the results of the DM care related investigations, the short-term outcome indicators could not be analyzed in our investigation.

Round 2
Reviewer 1 Report
Comments and Suggestions for Authors
I thank the authors for reviewing the manuscript.
I do not share some of your views, for example the primary care doctor does not usually supervise the work of the nurses, as they are competent and autonomous in their role of monitoring and providing health education to chronic patients with diabetes. In primary care, Interprofessional collaboration is fundamental to the integrative cooperation of different healthcare professionals, with complementary skills and making possible the best use of resources. (Samuelson M, al., 2012).
Registered nurses work in the Primary Care teams, have an autonomous role, including counseling people with type 2 diabetes in diabetes management and healthy lifestyle.
In relation to the analyzes based on the age of the doctors, I still disagree. Sorry, but the references you provide in your answer are from your environment and your team of researchers. Probably the most experienced primary care teams have professionals who are older. However this should not be a limitation to have the right skills to care for people. If so, the profile of these professionals should be studied and not just their age.
As for the rest of the changes made, I have no further comments.
Author Response
Dear Reviewer,
Thank you very much for the careful review of the revised version of our manuscript. Please find enclosed the 2nd revised version of the manuscript “Relationships between the structural characteristics of general medical practices and the socioeconomic status of patients with diabetes-related performance indicators in primary care” by Undraa Jargalsaikhan, et al.
Each comment and suggestion has been considered. The corresponding changes and refinements made in the revised paper are summarized in our response after considering each of your suggestion. Answers along with the modifications we made are summarized below (Your comments in capitals).
Sincerely yours, Janos Sandor (on behalf of the authors)
Answers/reflections to the comments of Reviewer-1:
1.
I THANK THE AUTHORS FOR REVIEWING THE MANUSCRIPT.
Thank you for this comment! We did our best.
2.
I DO NOT SHARE SOME OF YOUR VIEWS, FOR EXAMPLE THE PRIMARY CARE DOCTOR DOES NOT USUALLY SUPERVISE THE WORK OF THE NURSES, AS THEY ARE COMPETENT AND AUTONOMOUS IN THEIR ROLE OF MONITORING AND PROVIDING HEALTH EDUCATION TO CHRONIC PATIENTS WITH DIABETES. IN PRIMARY CARE, INTERPROFESSIONAL COLLABORATION IS FUNDAMENTAL TO THE INTEGRATIVE COOPERATION OF DIFFERENT HEALTHCARE PROFESSIONALS, WITH COMPLEMENTARY SKILLS AND MAKING POSSIBLE THE BEST USE OF RESOURCES. (SAMUELSON M, AL., 2012). REGISTERED NURSES WORK IN THE PRIMARY CARE TEAMS, HAVE AN AUTONOMOUS ROLE, INCLUDING COUNSELING PEOPLE WITH TYPE 2 DIABETES IN DIABETES MANAGEMENT AND HEALTHY LIFESTYLE.
We agree with you. The proper structure for DM care is multidisciplinary and involves nurses with DM license. Unfortunately, the Hungarian primary care has a more traditional structure (solo practice with one GP and one nurse). To declare it in the manuscript, the following sentences were added to the text:
The overwhelming majority of the Hungarian GPs work in their own solo practice with one nurse. Their service is free of charge for patients. Each GMP is monitored for performance by the NHIF. The DM monitoring at the level of primary care is of high importance in Hungary, because the DM care is mainly managed by the GP from the screening and diagnosis, through supporting of lifestyle rationalization, to the reference of patients for laboratory and clinical investigations.
3.
IN RELATION TO THE ANALYZES BASED ON THE AGE OF THE DOCTORS, I STILL DISAGREE. SORRY, BUT THE REFERENCES YOU PROVIDE IN YOUR ANSWER ARE FROM YOUR ENVIRONMENT AND YOUR TEAM OF RESEARCHERS. PROBABLY THE MOST EXPERIENCED PRIMARY CARE TEAMS HAVE PROFESSIONALS WHO ARE OLDER. HOWEVER THIS SHOULD NOT BE A LIMITATION TO HAVE THE RIGHT SKILLS TO CARE FOR PEOPLE. IF SO, THE PROFILE OF THESE PROFESSIONALS SHOULD BE STUDIED AND NOT JUST THEIR AGE.
One of the most important problem is the aging population of the GPs in many countries – Hungary is not an exception in this respect. An obvious consequence is that we have to know whether the GPs who works after reaching the retirement age can provide the care as effectively as the younger GPs. As it is shown in the bivariate analyses – added to the revised version of the manuscript – the GMPs with GP older than 65 provide the investigated elements of DM care less effectively. This is a descriptive finding without any intention of discrimination. The multivariate regression modeling confirmed the negative impact of the GPs’ age. The mechanisms by which the GP age above 65 influence the effectiveness of DM care needs further investigation, as it is explicitly acknowledged by the sentence added to the 4.2. Strengths and limitations section:
“More studies are needed to utilize the opportunities to improve DM care by improving the geographical availability in rural areas, by providing care more adapted to the less educated patients, by supporting the GMPs with bigger list size, and by understanding the negative impact of GPs’ age over 65 on process indicators.”
The cited papers demonstrating the importance of GPs’ age were published by our teams in Hungary. These were to show that the conclusion of our present manuscript is not an exceptional finding. But, we added a reference (#34) which dealt with the problem of aging physicians’ population in European context (the paper was published in BMJ). This paper emphasizes that the physicians’ aging is a characteristics of the health care we have to investigate (in order to understand and, if required, to mitigate the consequences). Because the GP age seems to be a factor that influence the effectiveness of the DM care, its investigation is the interest of patients. Therefore, its investigation cannot be considered as a discrimination towards the aged.
4.
AS FOR THE REST OF THE CHANGES MADE, I HAVE NO FURTHER COMMENTS.
Again, thank you very much for your careful review!

Reviewer 2 Report
Comments and Suggestions for Authors
Dear Authors:
Although I value very positively the revision made and I consider that the new manuscript that you have sent has improved notably, there are still aspects that, from my point of view, require further clarification.
I am sending these contributions in the attached document. I have removed from this document all the aspects that have already been resolved.
Best regards

Author Response
Dear Reviewer,
Thank you very much for the careful review of the revised version of our manuscript. Please find enclosed the 2nd revised version of the manuscript “Relationships between the structural characteristics of general medical practices and the socioeconomic status of patients with diabetes-related performance indicators in primary care” by Undraa Jargalsaikhan, et al.
Each comment and suggestion has been considered. The corresponding changes and refinements made in the revised paper are summarized in our response after considering each of your suggestion. Answers along with the modifications we made are summarized below (in italics).
Sincerely yours, Janos Sandor (on behalf of the authors)
Answers/reflections to the comments of Reviewer-2:
Page #1
3.
INFORMATION RELATED TO SOCIOECONOMIC STATUS AND ITS IMPACT ON DM NEEDS TO BE INCLUDED IN THE INTRODUCTION. THE CHARACTERISTICS OF GMP ARE NOT EXPLAINED EITHER.
- The introduction has been completed as requested to mention that there is an association between the socioeconomic position of patients and the epidemiological characteristics of DM.
Original text:
According to a recent Hungarian investigation, the quality of DM care improved glycemic control by 11.53%, and there was moderate improvement in diastolic blood pressure (6.26%), and total cholesterol (6.06%) therapeutic target achievement [19].
Modified text:
According to a recent Hungarian investigation, the quality of DM care improved in the respect of glycemic control (11.53%), diastolic blood pressure (6.26%), and total cholesterol (6.06%) therapeutic target achievement. This improvement was affected by the level of patients’ education. [19].
How socioeconomic factors affect DM remains to be described based on the literature.
The original text:
This improvement was affected by the level of patients’ education. [19].
The modified text:
This improvement was more pronounced among patients with higher level of education [19].
This improvement was more pronounced among patients with higher level of education.
- The role of GMP characteristics to affect the quality of the care was investigated. It was the declared objective to describe the association between GMP characteristics and DM indicators.
Some minimum characteristics to understand what general medical practice (GMP) consists of are still not described: Is care provided in community health centers, in private clinics, is it free of charge or do patients pay per visit, are nurses involved in the follow-up of diabetic patients…?
The original text:
The NHIF is the only entity in the country that covers the whole country, and each GMP is monitored for performance.
The modified text:
The overwhelming majority of the Hungarian GPs work in their own solo practice with one nurse. Their service is free of charge for patients. Each GMP is monitored for performance by the NHIF.
Page #2
5.
INFORMATION RELATED TO DIABETES CONTROL AND MANAGEMENT NEEDS TO BE INCLUDED. THE STUDY EXPLAINS WHAT DM INDICATORS ARE FOR BUT DO NOT EXPLAIN WHICH THESE ARE OR HOW DIABETES IS NORMALLY MANAGED AT THE CLINICS.
The criticized sentence is about the general aim of indicators’ applications, not about the details on the operation of DM indicators. (See response #48.)
It still does not describe what the usual care provided to diabetic patients is. In answer #48 nothing is said about this.
Original text:
The NHIF is the only entity in the country that covers the whole country, and each GMP is monitored for performance.
Modified text:
The NHIF is the only entity in Hungary that covers the whole country. The over-whelming majority of the Hungarian GPs work in their own solo practice with one nurse. Their service is free of charge for patients. Each GMP is monitored for performance by the NHIF. The DM monitoring at the level of primary care is of high importance in Hungary, because the DM care is mainly managed by the GP from the screening and diagnosis, through supporting of lifestyle rationalization, to the reference of patients for laboratory and clinical investigations.
6.
WHAT IS CONSIDERED QUALITY OF CARE?
The sentence had been reformulated. See the response #3a.
There is still no description of what is meant by quality of DM care. In answer #3, the impact of this care is described, but what does it consist of?
The term “quality” is not necessary in this sentence – and this manuscript was not devoted to the theoretical discussion of different aspects of the quality of DM care. The term was deleted.
Original text:
According to a recent Hungarian investigation, the quality of DM care improved in the respect of glycemic control (11.53%), diastolic blood pressure (6.26%), and total cholesterol (6.06%) therapeutic target achievement.
Modified text:
According to a recent investigation, the DM care in Hungary improved in the respect of glycemic control (11.53%), diastolic blood pressure (6.26%), and total cholesterol (6.06%) therapeutic target achievement.
9.
THE ORIGIN, TYPE AND CHARACTERISTICS OF THIS DATABASE MUST BE CLEARLY EXPLAINED. IS IT A HEALTH DATABASE? WHO HAS DEVELOPED IT? WHAT TYPE OF VARIABLES ARE INCLUDED?
Thanks for this comment! The National Health Insurance Fund is contracted with each GMPs in Hungary, and has data for the operation of each GMPs. To emphasize it the text was corrected.
Original text:
The data required for assessing indicators (patient characteristics and GMP structural parameters) were retrieved from the NHIF's integrated information system.
Modified text:
The data required for assessing indicators (patient characteristics and GMP structural parameters) were retrieved from the NHIF's integrated information system. NHIF is the national institution that contracts with each GMP of Hungary.
The characteristics of the database from which the data were obtained are still not described. Is it a health database? What type of variables are included?
Characteristics, which have been mentioned yet in the manuscript:
- nationwide covering each GMP
- health insurance database
- sole insurance company for primary care
- collecting data on processes not including outcome data
10.
IN THE DATABASE, DOES THE DIAGNOSIS OF DM NOT APPEAR? DID THEY NOT INCLUDE DM PATIENTS WHO ARE WITHOUT PHARMACOLOGICAL TREATMENT?
The original sentence:
„DM patients were defined using the operational definition of the NHIF by at least 4 redemptions of DM medicines in a year.”
Was completed with the suggested information:
„DM patients were defined using the operational definition of the NHIF by at least 4 redemptions of DM medicines in a year. DM patient s without pharmacological treatment were not included in the investigation.”
It remains unclear whether or not the database contains a diagnosis of DM.
The suspicion and the confirmed diagnosis is coded by the ICD-X code of DM. Therefore, the coded DM is not proper to identify the DM patients for the investigation. The suspected but not confirmed cases are sorted out from the target population by the drug consumption data.
12.
WHAT DOES THIS SYSTEM CONSIST OF AND HOW WAS THE INFORMATION RETRIEVED?
A new sentence to provide info on the National Health Insurance Fund was inserted.
Original text:
… retrieved from the NHIF's integrated information system
Modified text:
… retrieved from the NHIF's integrated information system. NHIF is the national institution that contracts with each GMP of Hungary.
The characteristics of the information system or database from which the data were obtained remain unclear.
See the response to the comment #9.
17.
THE MEANING OF THIS IS NOT WELL UNDERSTOOD
Carstairs score is a measure of deprivation for GMPs used in the UK.
Add in the text the following clarification
Original text:
The GMP-level SES indicators were based on census data as the Carstairs score components [23,24].
The modified text:
The GMP-level SES indicators were based on census data as the Carstairs index and the Townsend deprivation scores were computed in the UK [23,24].
19.
WHY WERE THESE AGE GROUPS CONSIDERED? THEY WOULD HAVE TO CLARIFY WHY THEY CONSIDER DISEASE PREVALENCE DATA AS INDICATORS OF QUALITY OF DM CARE.
The P4P system operated by the National Health Insurance Fund (mentioned in the Introduction, 3rd paragraph) uses prevalence indicators for hypertension. In this system, the prevalence of hypertension is calculated for two age groups: for younger adults with age of 40-54, and for older adults with age of 55-69. The same age groups were applied in our investigation.
It should be clarified in the text.
Original text:
the prevalence of DM patients among individuals aged 40–54 years and among individuals aged 55–69 years;
Modified text:
the prevalence of DM patients among individuals aged 40–54 years and among individuals aged 55–69 years (the same age groups are used in the NHIF P4P system to monitor the hypertension prevalence);
Page #4
20.
THE REASONS OF CHOOSING THESE AGE GROUPS CUT OFF POINTS NEEDS TO BE EXPLAINED
Adulthood is defined by 18 year of age in Hungary. We applied 5-year age groups. So first two years were defined a the first age group.
They should clarify in the text why they have selected age groups every 5 years. In principle, it seems too many groups. Is there any study that identifies differences in these very limited age groups?
Yes, the more detailed is the age categorization, the more precise the expected value.
21.
I UNDERSTAND THAT IT WILL BE GREATER THAN 90
Thanks for this comment! Modified accordingly.
Original text:
and 90<
Modified text:
and 90≤
It should be >90 and not <90
Thanks for this comment!
Original text:
by age group (18–19, 20–24, 25–29, 30–34, 35–39, 40–44, 45–49, 50–54, 55–59, 60–64, 70–74, 75–79, 80–84, 85–90, and 90<)
The modified text:
by age group (18–19, 20–24, 25–29, 30–34, 35–39, 40–44, 45–49, 50–54, 55–59, 60–64, 70–74, 75–79, 80–84, 85–89, and 90≤)
23.
EXPLAIN HOW THE ASSOCIATION BETWEEN SES AND DD IS CALCULATED.
I think that the process of the calculation is described step-by-step. Finally linear regression model is applied to quantify the relationship between SES and DM (not DD) indicators.
This should be added in the text. The statistical analysis section does not show how they calculate the relationship between SES and DM indicators.
The applied method was linear regression modelling, as it is mentioned in the “Multivariable linear regression models were applied to evaluate the relationship between GMP characteristics and transformed diabetes indicators (nISRs).” sentences of the 2.4. Statistical analysis section.
24.
BEING QUALITATIVE OUTCOME VARIABLES, WHY DIDN'T YOU USE LOGISTIC REGRESSION? WHY DIDN'T YOU PERFORM A BIVARIATE ANALYSIS BEFORE PERFORMING THE MULTIVARIATE ANALYSIS? WHY DIDN'T YOU USE THE EXPONENTS OF THE BETA COEFFICIENT (ODDS RATIO) WITH THEIR CIS, INSTEAD OF THE BETA COEFFICIENT?
- Linear model was used because the outcome were continuous parameters.
Ok
- Thanks for this suggestion! The bivariate linear regression analyses are summarized in Appendix Table A2, and a sentence was inserted into the end of 3.1. Descriptive Statistics section:
„The associations between GMP characteristics and nISRs by bivariate linear regression analyses are shown Appendix Table A2.”
Nothing is said about these results in the text.
Original text:
The associations between GMP characteristics and nISRs by bivariate linear regression analyses are shown Appendix Table A2.
Modified text:
According to the bivariate analyzes each indicator was reduced in GMPs with GP aged over 65. The process indicators was in inverse relationship with the list size. Similar associations were observed for low level of education and for rurality (apart from the influenza vaccination coverage). The prevalence of DM was lower among patients with urban residential place, higher level of education, being employed, and living in less crowded households. (Appendix Table A2)
Consequently, the numbering of the former Appendix Table A2 had to be changed to Appendix Table A3.
29.
SOME OF THE VARIABLES IN THE RESULTS AND THEIR CATEGORIES HAVE NOT BEEN EXPLAINED IN METHODS SECTION
List size categories are defined in the first sentence of the 2.2. Explanatory Variables section.
GMP type (adult or mixed) is not explained in the Methodology section.
Thanks for this comment!
Original text:
GMP structural characteristics were determined by the number of patients receiving GMP care (list size categories: ≤800, 801-1200, 1201-1600, 1601-2000, and ≥2001), status of GP vacancy (distinguishing GMPs managed by temporary GPs by limited time availability and permanent positions with contracted GPs with continuous availability), type of residence (urban, rural), age of GP (<65 or 65≥ years), and county of GMPs.
Modified text:
GMP structural characteristics were determined by the number of patients receiving GMP care (list size categories: ≤800, 801-1200, 1201-1600, 1601-2000, and ≥2001), status of GP vacancy (distinguishing GMPs managed by temporary GPs by limited time availability and permanent positions with contracted GPs with continuous availability), type of residence (urban, rural), age of GP (<65 or 65≥ years), type of GMP by patients provided (making distinction between GMPs for adults only and GMPs both for adults and for children as mixed) and county of GMPs.
Page #6
35.
THE VARIABLES AND THEIR CATEGORIES NEED TO BE INDICATED IN METHODS SECTION
Those are defined in the first paragraph in the 2.2. Explanatory Variables section.
GMP type adult mixed, and none of the patients characteristics (level of education, employment ratio, housing density...) are included in the 2.2. Explanatory Variables section
The explanation for GMP type has been added (see comment #29). The SES indicators are described in the following paragraphs of the “2.2. Explanatory Variables section”:
The number of years of school attendance among inhabitants aged at least 7 years was determined for each type of residence. The expected number of years of school attendance was calculated using national age- and sex-specific reference values and the demographic data of the residential areas. The indirectly standardized age- and sex-adjusted relative level of education (srEDU) was computed for areas of residence.
The settlement-specific employment ratio was standardized by age and sex to compute the standardized relative employment ratio (srEMP). The expected numbers for each area of residence were calculated using the age- and sex-specific national reference employment ratios (over the age of 15) and settlement data by demographic stratum.
The crowding index for each area of residence was calculated as the number of occupants over the number of rooms in the area. The average Hungarian crowding index was used to determine the residence-specific relative housing density (rHD).
The only significant ethnic minority of Hungary is the Roma. The relative Roma proportion (rRP) for an area of residence was defined as the proportion of self-declared Roma individuals in an area divided by the population share of self-declared Roma individuals in Hungary.
36.
CATEGORIES OF DIFFERENT VARIABLES ARE MIXED
The age of GP cannot be investigated if there is no GP contracted. Therefore, the vacancy and the age of GP have to be combined in one explanatory variable. As it is shown in the Table 3 the vacant GMPs and the GMPs with GP aged at least 65 are compared to the GMPs with GP aged less than 65.
Still not understood. GP (vacancy/age>65) could also be evaluated. If it is left it would be necessary to add the explanation in methodology.
For example, the comparison of GMP(vacancy) with GMP(age≥65) in the respect of the influenza vaccination can be done by checking the 95% confidence intervals of the adjusted regression coefficients (data from the Table 3):
GP (vacancy/ age<65): -0.053 [-0.083; -0.022]
GP (age≥65X/age<65): -0.032 [-0.060; -0.003]
Because of the overlapping between the 95% confidence intervals, there is no significant difference between GMP(vacancy) and GMP(age≥65) groups. But both is worse compared to the GMP(age<65).
page #7
42.
THE DISCUSSION ONLY COMPARES THE RESULTS OF THIS STUDY WITH OTHER RESEARCH, IT LACKS PERSONAL REFLECTION AS TO WHY OR WHAT COULD BE THE REASONS OF THESE RESULTS.
We did not investigate the determinants of the associations between patients and GMP characteristics and the DM care quality. So it could be not established to hypothesize the underlying causes.
Could you describe in the discussion why those factors might be influencing the results
We did not investigated the mechanisms by which the GMP characteristics influence the DM care, and we could not conclude regarding the causal mechanisms behind the observed associations. Therefore, the possible explanations can be added to the text as objectives for further investigations in the 4.2. Strengths and limitations section.
Original text:
Finally, the mechanisms behind the observed associations between GMP structural characteristics and DM care quality could not be investigated in our cross-sectional investigation.
Modified text:
Finally, the mechanisms behind the observed associations between GMP structural characteristics and DM care quality could not be investigated in our cross-sectional investigation. More studies are needed to utilize the opportunities to improve DM care by improving the geographical availability in rural areas, by providing care more adapted to the less educated patients, by supporting the GMPs with bigger list size, and by understanding the negative impact of GPs’ age over 65 on process indicators.
43.
THIS SET OF 7 INDICATORS HAS NOT BEEN EXPLAINED IN THE STUDY. MORE DATA RELATED TO WHY THESE 7 INDICATORS WERE CHOSEN NEEDS TO BE INCLUDED IN THE INTRODUCTION SECTION.
See response #48.
They should explain in the Introduction why these 7 indicators are chosen. In answer 48 you do not show this information
As it was mentioned in response #48 (and as the manuscript was completed accordingly) data for clinical status are not registered by the NHIF, so the short-term outcome indicators could not be evaluated. Regarding the process indicators: the EUBIROD process indicators were taken into consideration. Unfortunately, data for microalbuminuria and blood pressure assessment, and foot examinations are not registered properly in the NHIF system.
Original text:
The seven (two part of the routine monitoring system, five newly developed based on the data availability in NHIF) indicators assessed were …
Modified text:
The EUBIROD process indicators were taken into consideration. [17,18] Because data for microalbuminuria and blood pressure assessment, and foot examinations are not registered properly in the NHIF system, a restricted set of indicators could be investigated. The seven (two part of the routine monitoring system, five newly developed based on the data availability in NHIF) indicators assessed were …
44.
IT WOULD BE NECESSARY TO JUSTIFY WITH BIBLIOGRAPHY THAT THE CREATININE VALUE IS THE MOST IMPORTANT LABORATORY TEST IN DM CARE.
The primary alteration in DM is the impairment of the glycemic status. All organ impairments are complications caused by the impaired glycemic control. In this sense, the most important indicator is the indicator for glycemic control status.
It remains unclear why the creatinine value is the most important parameter in the control of DM.
Thanks for this comment! The sentence has been restructured.
Original text:
The GMP-level nISRs exhibited the narrowest distribution for the glycemic status evaluation indicator (the SD of the nISR was 0.082 for HbA1c measurement), followed by the lipid and serum creatinine levels (the SD of the nISR was 0.133 for lipids and 0.133 for serum creatinine), which is the most important laboratory examination in DM care.
Modified text:
The GMP-level nISRs exhibited the narrowest distribution for the glycemic status evaluation indicator (the SD of the nISR was 0.082 for HbA1c measurement) , which is the most important laboratory examination in DM care, followed by the lipid and serum creatinine levels (the SD of the nISR was 0.133 for lipids and 0.133 for serum creatinine).
45.
WHICH ONES?
As it is shown in the Table 3: education and employment.
Specify it in the discussion
Original text:
The proxy measures showed that the SES of patients had a greater impact than GMP structural parameters.
Modified text:
The proxy measures showed that the SES of patients (mainly education) had a greater impact than GMP structural parameters.
46.
NOTHING IS SAID ABOUT THESE TESTS IN THE STUDY.
The criticized sentence is to show that there are international publications on the similar association between DM care process indicators and the rurality of the GMP. That is there are similar observations published as we could observe in our investigation.
Their study did not evaluate aspirin therapy or microalbumin screening and therefore does not make sense to compare it with other studies.
The cited study found association between process indicators and rurality. We have similar observation. Although, the process indicators were not the same in the cited and in our study, the nature of the association was similar. Because of this similarity in conclusions, we think that it has importance that there is concordance between our observation and the former published experience.
Page #8
47.
MONOTONIC? THIS IS NOT WELL UNDERSTOOD
A monotonic function is a function between ordered sets that preserves or reverses the given order.
Still don't understand what this has to do with a "monotonic relationship".
There is 18,259 papers published with the “monotonic” term according to the PubMed (13/03/2024).
